

# Chemical Properties and Single Particle Mixing State of Soot Aerosol in Houston during the TRACER Campaign

Ryan N. Farley[1,2], James E. Lee[3], Laura-Hélèna Rivellini[4], Alex K. Y. Lee[5], Rachael Dal Porto[6],
Christopher D. Cappa[2,6], Kyle Gorkowski[3], Abu Sayeed Md Shawon[3], Katherine B. Benedict[3], Allison C. Aiken[3], Manvendra K. Dubey[3], Qi Zhang[1,2]

[1]Department of Environmental Toxicology, University of California Davis, CA, 95616, USA
[2]Agricultural and Environmental Chemistry Graduate Group, University of California Davis, CA, 95616, USA
[3]Earth and Environmental Sciences Division, Los Alamos National Laboratory, Los Alamos, NM, USA
[4] Department of Chemistry, University of Toronto, Toronto, ON, Canada
[5]Air Quality Processes Research Section, Environment and Climate Change Canada, Toronto, ON, Canada
[6]Department of Civil and Environmental Engineering, University of California, Davis, CA, 95616 USA

Correspondence to: Qi Zhang (dkwzhang@ucdavis.edu)

## Abstract

A high-resolution soot particle aerosol mass spectrometer (SP-AMS) was used to selectively measure refractory black carbon (rBC) and its associated coating material using both the ensemble size-resolved mass spectral mode and the event trigger single particle (ETSP) mode in Houston, Texas in summer 2022. This study was conducted as part of the Department of Energy Atmospheric Radiation Measurement (ARM) program's Tracking Aerosol Convection Interactions Experiment (TRACER) field campaign. The study revealed an average ($\pm 1\sigma$) rBC concentration of $103 \pm 176$ ng m$^{-3}$. Additionally, the coatings on the BC particles were primarily composed of organics (59%; $219 \pm 260$ ng m$^{-3}$) and sulfate (26%; $94 \pm 55$ ng m$^{-3}$). Positive matrix factorization (PMF) analysis of the ensemble mass spectra of BC-containing particles resolved four distinct types of soot aerosol, including an oxidized organic aerosol (OOA$_{BC,PMF}$) factor associated with processed primary organic aerosol, an inorganic sulfate factor (SO$_{4,BC,PMF}$), an oxidized rBC factor (O-BC$_{PMF}$), and a mixed mineral dust/biomass burning aerosol factor with significant contribution from potassium (K-BB$_{BC,PMF}$). Additionally, K-Means clustering analysis of the single particle mass spectra identified eight different clusters, including soot particles enriched in hydrocarbon like organic aerosol (HOA$_{BC,ETSP}$), sulfate (SO$_{4,BC,ETSP}$), two types of rBC, OOA (OOA$_{BC,ETSP}$), chloride (Cl$_{BC,ETSP}$) and nitrate (NO$_{3,BC,ETSP}$). The single particle measurements demonstrate substantial variation in BC coating thickness with coating-to-rBC mass ratios ranging from 0.1 to 100. The mixing state index ($\chi$), which denotes the degree of homogeneity of the soot aerosol, varied from 4 to 94% with a median of 40%, indicating that the aerosol population lies in between internal and external mixing but has large temporal and source type variability. In addition, a significant fraction of BC-containing particles, a majority enriched with oxidized organics and sulfate, exhibit sufficiently high $\kappa$ values and diameters conducive to activation as cloud nuclei



under atmospherically relevant supersaturation conditions. This finding bears significance in comprehending the activation of rBC-containing particles as cloud droplets and the origins of CCN in urban areas. Our analysis highlights the complex nature

of soot aerosol and underscore the need to comprehend its variability across different environments for accurate assessment of climate change.

## 1. Introduction

Soot aerosol, predominantly emitted from the incomplete combustion of fossil fuels and biomass, strongly absorbs solar radiation and contributes to positive climate forcing (Bond et al., 2013; IPCC, 2021; Ramanathan and Carmichael, 2008). The

microphysical and optical properties of soot aerosol are profoundly influenced by the mixing state of black carbon (BC) and aerosol constituents internally mixed with BC (Cappa et al., 2012). The properties of ambient soot aerosol undergo dynamic variations during their aging processes in the atmosphere, such as the condensation of low-volatility or semi-volatile vapors, coagulation with preexisting aerosol and heterogenous oxidation.

When freshly emitted, soot aerosol is often hydrophobic, but condensation of hygroscopic material, such as secondary organic

aerosol (SOA) or inorganic salts, enables soot to act as cloud condensation nuclei (CCN) and alter cloud properties (Lambe et al., 2015; Motos et al., 2019; Wu et al., 2019; Zuberi et al., 2005). Additionally, the rate at which BC is scavenged by wet deposition, and therefore its atmospheric lifetime, is controlled by the hygroscopicity of its coating (Emerson et al., 2018; Yang et al., 2019). The presence of non-absorbing coating materials can also enhance aerosol light absorption through the so-called lensing effect, leading to disparities between optical models and observations (Cappa et al., 2012, 2019; Fierce et al.,

2020; Lack and Cappa, 2010; Lee et al., 2022; Xie et al., 2019). The competition between the absorption enhancement and reduced lifetime from internal mixing of BC significantly effects its climate impacts and is not well understood (Hodnebrog et al., 2014).

Understanding the chemical and microphysical properties of soot aerosol and the complexities associated with its mixing state, which involves the interaction and incorporation of BC with other aerosol components, is crucial for comprehending its impact

on the climate system and refining climate change assessment. Process level modelling studies with detailed mixing state treatments show that current simple model parameterizations can bias the climate forcing of BC (Fierce et al., 2017)

To address this problem, the present study investigates the intricate properties and dynamic variations of mixed soot particles with state-of-the-art chemical and microphysical instruments to determine their influence on cloud development in Houston, TX. This study is part of the DOE Sponsored TRacking Aerosol Convection Interactions Experiment (TRACER), which aims

to unravel the interactions between clouds and aerosols in urban areas. Houston is a subtropical city heavily impacted by anthropogenic emissions from gasoline and diesel vehicles, cargo ships, and petrochemical industries (Al-Naiema et al., 2018; Bean et al., 2016; Schulze et al., 2018; Yoon et al., 2020). The abundance of soot particle sources and the high levels of photochemical activity in Houston make it an ideal location for studying urban soot aerosol characteristics and atmospheric processing.



Previous ambient measurements in Houston have reported average BC concentrations ranging between 0.31 and 0.80 µg m$^{-3}$ (Levy et al., 2013; Yoon et al., 2020).  Additionally, a study utilizing ambient vapors within an environmental chamber in Houston found that after 18 hours of atmospheric processing, monodisperse BC aerosol is transformed from a highly fractal morphology into spherical particles with a significant simultaneous increase in absorption enhancement ($E_{abs}$) (Peng et al., 2016). Another study by Levy et al. (2013) utilized measurements of BC aerosol effective density to probe the mixing state of

soot particles in Houston and found that although the soot particles were often internally mixed with other species, an externally mixed BC population was also present during rush hour.

Despite the importance of soot aerosol at this site, detailed measurements of the composition and properties of BC-containing aerosol have not been performed in Houston.  A particularly significant gap in knowledge pertains to the mixing states of soot aerosols. In this study, a soot particle aerosol mass spectrometers (SP-AMS) configured with only the laser vaporizer was used

to selectively measure submicrometer particles (PM$_1$) containing black carbon.  Additionally, a co-located dual-vaporizer SP-AMS, equipped with a PM$_{2.5}$ inlet was deployed concurrently to provide measurements of the bulk PM$_{2.5}$ composition. Comprehensive measurements of aerosol optical properties were also collected to explore the effects of mixing state and coating composition on light absorption. Here, we report the results of ensemble and single particle measurements of BC-containing aerosol and use them understand how size resolved mixing leads to compositional differences between BC-

containing and BC-free aerosols.

## 2. Methods

### 2.1    Sampling Location and Instrumentation

Measurements were performed at a U.S. Department of Energy (DOE) managed site at the La Porte Municipal Airport (29.670, -95.058) from Jun 28 to Aug 1, 2022, as a part of the TRACER intensive operating period.  The La Porte site is located

approximately 30km ESE of downtown Houston and is surrounded by residential and commercial areas (Fig. S1).  The Port of Houston and Houston Ship Channel (HSP) are located about 5 km to the northeast and there are multiple refineries and other industrial and petrochemical plants located in the region.  Although there was an active small aircraft runway nearby, we see no evidence of aircraft emissions impacting our measurements.

A suite of instrumentation for the comprehensive characterization of aerosol chemical and optical properties was housed in a

temperature-controlled trailer owned by the Los Alamos National Laboratory (Fig. S1). Among them, two high-resolution time-of-flight soot particle aerosol mass spectrometers (SP-AMS; Aerodyne Inc., Billerica MA.) were deployed simultaneously.  One SP-AMS was operated in the laser-only configuration, allowing for the measurement of BC-containing PM$_1$, while the other SP-AMS incorporated both a laser and thermal vaporizer and a PM$_{2.5}$ aerodynamic lens system, aiming to characterize both BC-containing and non-refractory PM$_{2.5}$ (NR-PM$_{2.5}$) (Avery et al., 2020; Williams et al., 2013).  Details

of supporting instrumentation is provided in section S1.1



## 2.2 Soot Particle Aerosol Mass Spectrometer (SP-AMS) Operation and Data Analysis

### 2.2.1. SP-AMS measurement and data processing

The laser-only SP-AMS utilized in this study was modified by removing the tungsten thermal vaporizer in order to selectively detect particles containing black carbon. The SP-AMS is described in detail in previous publications (DeCarlo et al., 2006; Onasch et al., 2012). Briefly, ambient aerosols are sampled through an aerodynamic lens system and sorted based on their vacuum aerodynamic diameter ($D_{va}$) within the Particle Time of Flight (PToF) chamber. A Nd-YAG intracavity (1064nm) laser is utilized to selectively vaporize particles containing absorbing material, such as rBC. The resulting gaseous molecules are ionized using 70 eV electron impact ionization and measured using high resolution time of flight mass spectrometry. The mass spectrometer was operated in the "V" mode, providing higher sensitivity at the cost of lower resolution ($m/\Delta m = 2500$) and was programmed to switch between three different sampling modes: the ensemble mass spectrum (MS) mode, the efficient particle time of flight (ePToF) size mode, and the event trigger single particle (ETSP) mode (DeCarlo et al., 2006; Onasch et al., 2012).

MS and ePToF measurements were analyzed in the SQUIRREL (v. 1.65C) and PIKA (v. 1.25C) analysis toolkits within the IGOR Pro (v. 8.04) environment and concentrations were determined from the high-resolution peak fittings. Ambient rBC was quantified using the sum of $C_1^+$ to $C_{10}^+$. As the signal at $C_1^+$ can be influenced by both organics and rBC, the $C_1^+$ related to rBC was constrained using the ratio of $C_3^+$ to $C_1^+$ measured during the regal black calibration (0.65). The limit of detection (LOD) for each species was determined as $3\sigma$ during filtered air measurements when a HEPA filter was placed upstream of the SP-AMS. The LOD for 5 min averaging was 106, 17, 6, 11, 12, 2 ng m$^{-3}$ for organics, rBC, sulfate, nitrate, chloride and ammonium, respectively. Additional details of SP-AMS operation and calibration is provided in section S1.2.

### 2.2.2 Positive matrix factorization analysis of ensemble mass spectra

Source apportionment of the ensemble mass spectra was performed by positive matrix factorization (PMF) analysis using the PMF evaluation tool (PET) (Paatero and Tapper, 1994; Ulbrich et al., 2009). Organic and inorganic ions were included in the PMF matrix to reduce rotational ambiguity as well as to explore the distribution of inorganic species across the factors (Sun et al., 2012). This included HR organic ions between 12-120 amu, major ions related to rBC, sulfate and nitrate and UMR signal between $m/z$ 121-307. PMF Solutions up to 8 factors were assessed based on mass spectral and temporal features. Four meaningful factors were identified and includes an oxidized organic aerosol factor (OOA$_{BC,PMF}$), an oxidized black carbon factor (O-BC$_{PMF}$), an inorganic sulfate factor (SO$_{4,BC,PMF}$), and a mixed biomass burning/mineral dust factor (K-BB$_{BC,PMF}$).

### 2.2.3 Event trigger single particle (ETSP) data processing and K-means clustering

The Event Trigger Single Particle mode was used to explore the mixing state of individual ambient particles, as done previously (Lee et al., 2019; Ma et al., 2023; Willis et al., 2019; Ye et al., 2018). The vaporization, ionization, and detection processes are identical to what is described above, however rather than averaging the measured signals across all MS extractions,



individual extractions (single particle events) are identified and saved if the signal at predefined *m/z* values is above a set threshold. Table S1 shows the three regions of interest (ROIs), consisting of individual or a range of *m/z* values used to trigger particle events and the corresponding ion thresholds needed at each ROI. *m/z* 36 and 43 were selected to target rBC and OA,

respectively. ROI 3 encompassed the total signal within the range of 46-150 amu, allowing for capturing particle events containing sulfate, nitrate and/or high molecular weight organics. Note that the settings were altered slightly after the first week to better target OA.

The unit mass resolution (UMR) data was processed in Tofware v2.5.13 prior to inputting into the Cluster Input Preparation Panel (CIPP), a tool developed by Lee et al. (2015), to identify real particle events. For each event, the total ion signal was

defined as the sum of all ions except for *m/z* 14 ($N^+$), 16 ($O^+$), 18 ($H_2O^+$ and $^{18}O^+$), 20 ($Ar^{2+}$) 28 ($N_2^+$), 32 ($O_2^+$), and 40 ($Ar^+$), as they are significantly influenced by air signal. A simplified fragmentation table using the same fragmentation ratios as those from the ensemble measurements (Allan et al., 2004) was used to quantify the contribution of different aerosol species. The minimum ion threshold was set as the average number of ions plus three standard deviations in the particle-free regions (defined as 10-50 nm and 2000-4000 nm), and any events falling below this threshold were discarded. Throughout the campaign,

organized into 1-week segments, the minimum ion threshold varied slightly, ranging between 21-27 ions. Additionally, events were considered valid if the signal of any individual species (rBC, sulfate, nitrate, chloride, potassium) exceeded the average plus three standard deviations of that species within the particle-free region.

In total, 14,699 single particle events were identified and imported for K-means clustering. Clustering was performed within Igor Pro v. 8.04 using the Cluster Analysis Panel (CAP) (Lee et al., 2015). Prior to clustering analysis, additional, noisy ions

were removed including *m/z* 15 ($^{15}N^+$), 17 ($NH_3^+$, $HO^+$), 19 ($H_2^{18}O^+$), 22 ($CO_2^{2+}$), 182 ($^{182}W^+$), 183 ($^{183}W^+$) and 186 ($^{186}W^+$). Single particle spectra were normalized to the total ion signal. Solutions up to 15 clusters were analyzed based on change in Euclidian distance as well as the spectral features, size distributions and temporal patterns of each class (Figure S2).

The chosen solution included 12 classes, however clusters that showed similar features were merged for a more coherent representation (Fig. S3). Specifically, sulfate signal was found to be split between three different clusters: one predominantly

comprising *m/z* 64 ($SO_2^+$), another featuring elevated *m/z* 48 ($SO^+$), and a third with elevated *m/z* 80 ($SO_3^+$) and 81 ($HSO_3^+$). Although these spectra may potentially hold intrinsic significance (e.g., aerosol acidity), it is more likely that this splitting behavior captures statistical variability in sulfate fragmentation. Consequently, these clusters were combined. Two OOA-like clusters were also identified, showing comparable size distributions and temporal trends. However, due to limitations in the UMR signal, it was not possible to discern meaningful differences between these clusters and they were combined. Finally,

one cluster displayed signal spikes at seemingly random masses (*m/z* 76, 112, 130, 145), making it challenging to provide a physically meaningful interpretation. Given the limited occurrence of these events (< 0.5% of particle count), these spectra were assumed to be noise and were discarded from further analysis. $NH_x^+$ ions were not included in the particle clustering analysis; therefore, ammonium mass associated with each cluster was estimated assuming particle neutralization.





**2.3 Quantification of Aerosol Mixing State and Estimation of Single Particle Hygroscopicity**

160 The aerosol mixing state was quantified using the metrics presented in Riemer and West (2013). The single particle diversity ($D_i$) describes the number of chemical components in a single particle, weighted by the mass fraction of each component. The components used in this analysis were Org, rBC, sulfate, nitrate, chloride, and potassium. $D_i$ varies from 1 when a particle is composed of only a single component, to the total number of species (in this work 6) when every species composes equal mass fractions. $D_i$ is calculated using the following formula:

165 $$D_i = \prod_{a=1}^{A}(p_i^a)^{-p_i^a} \tag{2}$$

Where $p_i^a$ is the mass fraction of species $a$ within particle $i$ measured in the ETSP mode. The average single particle diversity ($D_\alpha$) is the weighted average of $D_i$ across an aerosol population and is calculated as:

$$D_\alpha = \prod_{i=1}^{N}(D_i)^{p_i} \tag{3}$$

Where $p_i$ is the mass fraction of particle $i$ in the population. $D_\gamma$ measures the bulk population diversity and is calculated as:

170 $$D_\gamma = \prod_{a=1}^{A}(p^a)^{-p^a} \tag{4}$$

Where $p^a$ is the mass fraction of species $a$ in the aerosol population measured in the ensemble mode.

While $D_i$ and $D_\alpha$ were calculated based on the single particle spectra from ETSP measurements, $D_\gamma$ is a parameter derived from the ensemble measurements. As the ensemble MS and ETSP measurements were carried out sequentially at 20 min interval, hourly averages of $D_\alpha$ and $D_\gamma$ were calculated to determine the mixing state index ($\chi$). $\chi$ describes the degree of population

175 homogeneity and varies from 0%, signifying a fully external mixture, to 100%, which indicates a fully internal mixture and is defined as:

$$\chi = \frac{D_\alpha - 1}{D_\gamma - 1} \tag{5}$$

The hygroscopicity of individual particles was predicted using Zdanovskii-Stokes-Robinson (ZSR) mixing rule and the hygroscopicity parameter ($\kappa$) introduced in Petters and Kreidenweis (2007). Details of these calculations is given in sections

180 S1.4 and S1.5.

**3. Results**

**3.1 Overview of Soot Aerosol Composition and Properties in Houston during TRACER**

**3.1.1 soot aerosol composition and diurnal variations**

Meteorological conditions during TRACER were hot and humid, typical of summertime conditions at this subtropical site

185 (Fig. S4). The average ($\pm 1\sigma$) temperature was 29 ± 3°C and average RH was 72 ± 13% with strong diurnal variation. Throughout the measurement period there were several heavy precipitation events, however these rarely lasted longer than a few hours. The wind showed a consistent diurnal profile with weak (< 2 m/s) southerly/southwesterly winds overnight and stronger (> 5 m/s) southeasterly winds in the afternoon (Fig. S5). This pattern is a result of a strong sea-breeze dynamic



previously identified in the Houston area and results in the accumulation of local emissions overnight, followed by the

advection of marine or processed airmasses during the day (Banta et al., 2005; Caicedo et al., 2019; Li et al., 2020).

The average black carbon concentration was $0.10 \pm 0.18$ µg m$^{-3}$, which is significantly lower than the average of $0.31 \pm 0.22$ µg m$^{-3}$ measured in May 2009 (Levy et al., 2013) and $0.80 \pm 0.69$ µg m$^{-3}$ measured in September 2013 (Yoon et al., 2020). However, both of those prior measurements were conducted at the University of Houston, which is closer to the urban core of Houston.  The rBC concentration measured by the laser-only SP-AMS agrees well with a co-located SP2 (Slope = 1.05, r$^2$ =

0.67, Figure 1e) and the dual-vaporizer SP-AMS with a PM$_{2.5}$ inlet (Slope = 1.14, r$^2$ = 0.62; Fig. S6).  The slightly higher rBC concentration measured by the dual vaporizer SP-AMS may be due to additional rBC mass present in the $1 - 2.5$ µm range. Additionally, the relationship between the SP-AMS rBC and the PM$_{2.5}$ aerosol absorption at 532nm, determined using orthogonal distance fitting constrained to an intercept of 0, yields a slope of 13.8 m$^2$ g$^{-1}$ (r$^2$ = 0.46).  This value is higher than the expected MAC of pure black carbon of $7.5 \pm 1.2$ m$^2$ g$^{-1}$ (Bond and Bergstrom, 2006; Liu et al., 2020), and can be explained

by absorption from dust particles, absorbing material present between $1 - 2.5$ µm or absorption enhancement due to BC mixing state (Cappa et al., 2012; Lack and Cappa, 2010).  Further analysis of the aerosol optical properties will be included in a separate publication.

The diurnal pattern of rBC concentration shows a weak bimodal cycle, with higher concentrations seen in the morning and early afternoon and lower concentrations in the evening and overnight (Fig. 1a).  The morning peak at 08:00 local time

coincides with high gas-phase NO$_x$ and CO concentrations and is likely related to fresh vehicle emissions during rush hour (Fig. 1a, S5).  Due to the high boundary layer height in the afternoon (2-3km), the increase of rBC between 16:00-17:00 is likely due to a combination of elevated vehicle emissions, industrial emissions, and the regional transport of BC-containing aerosol.

The soot aerosol coating thickness was estimated using the parameter R$_{Coat/BC}$, calculated by taking the mass ratio of coating

material to refractory black carbon.  The term "coating material" is used to refer to the total inorganic and organic species found within the soot particles and does not imply knowledge of particle morphology or the coating being present as a layer on the black carbon core. Figure 2a presents the frequency of R$_{coat/BC}$ occurrences for the entire campaign.  Each ensemble R$_{Coat/BC}$ measurement represents the average mass ratio of all submicrometer soot particles sampled during the five-minute averaging period, while the R$_{coat/BC}$ calculated from the single particle measurements represents the mass ratios for individual

soot particles.  The frequency histograms of R$_{coat/BC}$ display log-normal distributions for both the ensemble and single particle measurements, but with substantially different median values of 4.5 and 1.5, respectively (Fig. 2a).  The ensemble R$_{coat/BC}$ measurements agree well with previous measurements in urban areas, which typically vary between 1-10 (Collier et al., 2018; Healy et al., 2015; Wang et al., 2020a).  However, there were periods of thickly coated particles with R$_{Coat/rBC}$ values exceeding 10, indicating that a portion of the soot aerosol at this site may have undergone extensive atmospheric processing.  The

distribution of R$_{coat/BC}$ measured by ETSP highlights a population of nearly uncoated BC particles with R$_{coat/BC}$ less than 1, indicating substantial variation in the rBC mixing state.  The diurnal profile of ensemble R$_{Coat/BC}$ appears inversely related to the rBC concentration, with the lowest values occurring in the morning and afternoon (Fig 1a).  These time intervals are likely



influenced by fresh, local emissions from vehicles or industrial sources, showing similar $R_{Coat/BC}$ values to previous measurements taken near roadways (Lee et al., 2017). The soot aerosol coating is dominated by organic material (59%)

followed by sulfate (25%) and ammonium (9%) (Fig. 3a). Potassium contributes 3% while nitrate and chloride each 1%. Figure 2 also shows the chemical composition of rBC coatings at different $R_{coat/BC}$ values, determined from both ensemble spectra (Fig. 2c) and single particle spectra (Fig. 2d). Overall, as the $R_{coat/BC}$ ratio increases, the coating composition remains similar with a slight increase in the sulfate mass fraction, which is consistent with the formation of secondary aerosol upon atmospheric aging. Further details regarding the single particle composition across varying $R_{coat/BC}$ values, along with insights

into particle mixing states, are provided in the subsequent sections.

The diurnal cycle of $Org_{BC}$ is similar to that of rBC and shows a slight morning increase (Fig. 1b). The diurnal profiles of both rBC and $Org_{BC}$ show minima between 19:00 and 23:00, indicating the transport of cleaner, marine airmasses to the site. The $SO_{4,BC}$ profile shows little diurnal variation, highlighting its regional sources (Fig 1c). The ratio of $SO_{4,BC}$ to rBC peaks between 20:00 and 23:00 possibly due to long-range transport of sulfate-rich aerosol. The observed $SO_{4,BC}$ concentration was

unlikely to be significantly impacted by sea salt emissions, as the SP-AMS signals for $NaCl^+$, $Cl^+$ and $HCl^+$ were low despite these ions being detectable with the laser vaporizer if internally mixed with rBC. Instead, we hypothesize that $SO_{4,BC}$ is primarily attributed to ammonium sulfate formed from the oxidation of either marine-derived DMS or $SO_2$ from anthropogenic sources such as crude oil processing or other petrochemical industries (Ahmed et al., 2021; Rivera et al., 2010). We see little evidence for other forms of sulfate, such as methanesulfonic acid (MSA), primarily due to the absence of marker ions for

MSA, such as $CH_3SO_2^+$ and $CH_4SO_3^+$ (Ge et al., 2012a).

Figure 1b-d also shows the diurnal profiles of non-refractory $PM_{2.5}$ (NR-$PM_{2.5}$) collected by the collocated dual vaporizer SP-AMS in laser-off mode (Section S1.1). The comparison of each species between the laser-only SP-AMS and the laser-off SP-AMS provides insights into the percent of mass internally mixed with rBC. This assumes that the mass of refractory organics, sulfate, nitrate and chloride material internally mixed with BC is low, and that negligible non-refractory mass is present

between 1-2.5 µm. On average, approximately 27% of organic mass during this study is internally mixed with rBC, however, this fraction exhibits considerable temporal variation, as indicated by the moderate $r^2$ of 0.49 (Fig. 1f). Overall, both $Org_{BC}$ and $Org_{NR-PM2.5}$ show similar diurnal profiles (Fig 1b) however a slightly higher proportion of organic mass is mixed with rBC at night and during the morning hours, with the fraction decreasing in the afternoon. This pattern is consistent with vehicles and industrial sources emitting an internal mixture of rBC and organics, while SOA production tends to enhance the growth

of BC-free aerosols due to their higher surface area fraction. Supporting this hypothesis, only 3% of sulfate mass is mixed with rBC, despite a stronger correlation between $SO_{4,BC}$ and non-refractory $SO_4$ ($r^2 = 0.79$; Fig. 1g). As sulfate is primarily formed through secondary processes in the atmosphere, this limited presence of sulfate on soot particles indicates that the bulk of sulfate is externally mixed from rBC. Like SOA, the condensation of sulfate is likely to occur on all particles, however the higher particle surface area of non-BC-containing aerosol will increase the fraction externally mixed from rBC. Furthermore,

aerosol liquid water content is likely to be higher on particles free of rBC due to increased hygroscopicity, which promotes the aqueous phase formation of sulfate on non-BC-containing particles.



Nitrate, another secondary aerosol species, presents at low concentrations within both the rBC fraction and the NR-PM$_{2.5}$ fraction. High ambient temperatures throughout the measurement period likely promote the partitioning of inorganic nitrate into the gas-phase. The negligible correlation between NO$_{3,BC}$ and NO$_{3,NR\text{-}PM2.5}$, coupled with their different diurnal patterns

(Fig. 1h), highlights the overall externally mixed nature of nitrate and soot aerosol, and suggests different sources of nitrate between the two fractions. Additionally, differences in the aerosol water content or acidity between BC-containing and BC-free particles could also play a role in the weak correlation.

### 3.1.2 Single-particle characteristics of soot aerosol

Using the SP-AMS ETSP mode, we were able to further probe the single-particle composition and mixing state of BC-

containing aerosol. Although the 14699 single particle spectra recorded only represent a small subset of ambient BC-containing aerosol, a comparison of the average composition, mass spectral features and species-dependent size distributions measured during ETSP mode and ensemble mode indicates that the sampled single particles are generally representative of the soot population (Fig. 3). The mass fraction of rBC in PM$_{1,BC}$ was significantly higher in the single-particle measurements compared to the ensemble measurements, accounting for 36% of PM$_{1,BC}$ in comparison to 22% for the ensemble measurements.

Concurrently, the SO$_{4,BC}$ mass fraction decreased from 20% measured during ensemble mode to 13% during ETSP mode. These differences could potentially arise from biases created by the specific $m/z$ triggers selected for this study as has been observed previously (Lee et al., 2019). For example, one of the three triggers used was $m/z$ 36 (C$_3^+$), which may introduce a bias towards events with high $m/z$ 36 signal. Differences also appear when comparing the average mass spectra of all ensemble measurements (Fig. 3c) and all single particle events (Fig. 3d) which likely arise due to variations between the HR and UMR

fragmentation tables (Allan et al., 2004), in addition to the biases discussed above.

Despite these differences, the species dependent size distributions for SO$_{4,BC}$, Org$_{BC}$, rBC and K$_{BC}$ are similar between the bulk ensemble and ETSP measurements suggesting that there is minimal systemic bias as a function of particle size (Fig. 3). However, the ETSP size distribution for SO$_{4,BC}$ and Org$_{BC}$ are slightly shifted towards larger particle sizes. Additionally, the ensemble rBC size distribution shows an additional mode at small sizes. These differences may be attributed to larger particles

generating more ions and thereby having a higher statistical likelihood of triggering an ETSP event.

Although rBC or other absorbing material is required for vaporization and subsequent detection with the laser-only SP-AMS configuration, 49% of the ETSP spectra measured had no detectable rBC. The fraction of particles with no rBC was highest for the particle types with thickest coatings including HOA$_{BC,ETSP}$ (62%), SO$_{4,BC,ETSP}$ (45%) and OOA $_{BC,ETSP}$ (35%) particle clusters (Table S3). These seemingly rBC-free particles may represent events in which the particles did not overlap well with

the laser vaporizer, resulting in temperatures that were sufficient to vaporize non-refractory coating material but not the rBC core (Onasch et al., 2012; Willis et al., 2014). Another possibility is that these particles were extremely thickly coated with only a small BC core, and the energy supplied by the laser vaporizer was only sufficient to vaporize the coating material, leaving the rBC intact. Finally, semi-volatile, BC-free particles may be thermally vaporized on heated internal surfaces, such as the ion filament, despite the absence of the tungsten vaporizer. This could be especially relevant for the NO$_{3,BC,ETSP}$ cluster



due to the high volatility of ammonium nitrate. However, it is important to note that this cluster comprises only 247 particles, representing only 1.6% of the total single particles detected (Fig. 5g). Particles without rBC signal were included for the clustering analysis, but necessarily excluded from the calculation of $R_{coat/BC}$.

## 3.2 Sources and Atmospheric Processing of Soot Aerosol

### 3.2.1 Source apportionment using ensemble measurements

PMF analysis of the ensemble high-resolution mass spectra of soot particles resolved four distinct soot particle types characterized by distinct spectral features and tracer ions. These types include an oxidized organic aerosol ($OOA_{BC,PMF}$) factor, an oxidized rBC-rich factor ($O\text{-}BC_{PMF}$), a sulfate-rich factor ($SO_{4,BC,PMF}$), and a mixed biomass burning/mineral dust factor ($BB\text{-}K_{BC,PMF}$).

The $OOA_{BC,PMF}$ factor has an average concentration of $0.10 \pm 0.19$ µg m$^{-3}$, constituting 29% of the $PM_{1,BC}$ mass (Fig. 4). This soot type is primarily composed of organic species, which contribute 79% of the mass, with the rest of the mass comprised of rBC (14%), nitrate (4%) and sulfate (3%). The organic fraction is moderately oxidized with an O/C of 0.68 and shows enhanced levels of $C_2H_3O^+$ and $CO_2^+$, two markers for SOA (Ng et al., 2011). However, there is also an enhancement of $C_xH_y^+$ fragments, including $C_3H_3^+$ (*m/z* 39.02), $C_3H_5^+$ (m/z 41.04) $C_4H_7^+$ (*m/z* 55.05), $C_4H_9^+$ (*m/z* 57.07), which are frequently used as markers for hydrocarbon like OA (HOA) (Al-Naiema et al., 2018; Ge et al., 2012b; Young et al., 2016). Based on this, we conclude that this factor likely represents processed soot aerosol with coatings containing a mixture of SOA and oxidized primary OA. Additionally, 80% of the nitrate signal is attributed to this factor. The concentration of this factor peaks in the afternoon, consistent with secondary photochemical production (Fig. S7). However, there is also a minor increase corresponding with morning rush hour, emphasizing the contribution of primary aerosol (Fig. S7a). The time series shows the highest concentrations on Jun 28$^{th}$ – Jun 29$^{th}$ during which back trajectories are related to long range transport from the continental U.S (Fig. S8).

The $O\text{-}BC_{PMF}$ spectrum is dominated by the $C_{1-3}^+$ peaks, with the rBC and organic fragments contributing 82% and 17% of the mass respectively. The most abundant organic fragment is $CO_2^+$ and the spectrum also shows an enhancement of the $C_3O_2^+$ ion (*m/z* 67.99). These ions serve as a marker for oxidized functional groups on the soot surface (Falk et al., 2021; Ma et al., 2023). Elevated $C_3O_2^+$ and $CO_2^+$ signals have also been identified in POA emitted from diesel and gasoline vehicles and in soot particles emitted from wood or fossil fuel combustion (Carbone et al., 2019; Collier et al., 2015; Corbin et al., 2014, 2015). There is also slightly enhanced signal from reduced hydrocarbons such as $C_4H_7^+$ (*m/z* 55.05) and $C_4H_9^+$ (*m/z* 57.07), indicating that traffic emissions may be a source of this particle type. Indeed, the diurnal profile shows a clear increase at 7:00 and a minor increase at 16:00 local time, consistent with typical rush hours (Fig. S7b). Based on these temporal features, we conclude that this factor likely represents emissions from both gasoline light duty vehicles and industrial or diesel vehicle sources. The average concentration of $O\text{-}BC_{PMF}$ was $0.102 \pm 0.193$ µg m$^{-3}$, however, sporadic spikes were also seen,



with concentrations occasionally reaching 10 µg m$^{-3}$. The most intense of these instances occurred on Jul 7$^{th}$, coinciding with a plume observed near the site, however the fuel type for this event is unknown (Fig. 4b).

The SO$_{4,BC,PMF}$ factor is dominated by H$_y$SO$_x^+$ ions which contribute 67% of the factor mass, along with relatively minor contribution from rBC (23%) and organics (9%). The time series shows a consistent background presence of this factor
throughout the measurement period, with an average loading of 0.117 ± 0.071 µg m$^{-3}$ (31% of PM$_{1,BC}$; Fig. 4c). The low organic content suggests that this factor is mainly comprised of inorganic sulfate that has condensed onto ambient rBC particles. Overall, SO$_{4,BC,PMF}$ accounts for over 90% of the measured sulfate mass within soot aerosol. Rather than being associated with a specific source region, this factor appears to be a component of the regional background and is likely influenced by offshore sources (Fig. S8d).
An important source of non-sea salt sulfate in coastal regions is the oxidation of dimethyl sulfide (DMS), a byproduct from phytoplankton activity (Andreae, 1990; Andreae et al., 1985; Bates et al., 1992). Additionally, Bates et al. (2008) estimated that marine vessel emissions were the dominant source of sulfate over the Gulf of Mexico in 2006. However, recent regulations implemented in 2020 limit the sulfur content in marine heavy fuel oil to 0.1% by mass within the North American Emissions Control Area (ECA) and 0.5% in international waters (IMO, 2020). These policies have led to a noteworthy 80%
reduction in SO$_x$ emissions and 72% reduction in PM$_1$ emissions associated with shipping (Anastasopolos et al., 2021; Watson-Parris et al., 2022; Yu et al., 2020). Consequently, marine vessel emissions may contribute a smaller amount of sulfate and PM in this study compared to previous campaigns. Further discussion regarding the identification of shipping emissions is provided in section 3.3.

The fourth factor, denoted as BB-K$_{BC,PMF}$, is notable for a significant contribution from K$^+$ and accounts for 95% of
the total K$^+$ mass observed within soot aerosol. The average loading of BB-K$_{BC,PMF}$ was 0.058 ± 0.059 µg m$^{-3}$ with potassium contributing 7% of the total mass. The remaining mass consists of organic (59%), rBC (29%) and sulfate (5%). We propose that this factor represents a composite of biomass burning (BB) soot aerosol intermixed with mineral dust. Several key indicators support this hypothesis. K$^+$ is often used as a molecular marker for biomass combustion processes (Andreae, 1983) and the BB-K$_{BC,PMF}$ spectrum shows a slight increase in C$_2$H$_4$O$_2^+$ (m/z 60) and C$_3$H$_5$O$_2^+$ (m/z 73), which are markers for
anhydrous sugars such as levoglucosan that are commonly emitted during BB processes (Cubison et al., 2011). However, the K$^+$ time series also shows multi-day events between Jul 16$^{th}$ and Jul 23$^{rd}$ which coincide with enhancements of coarse-mode aerosol volume (V$_{P,1-10µm}$) and aerosol light absorption. These events are likely related to Saharan mineral dust transported from West Africa (Fig S9a, Gorkowski et al., in Prep). The presence of Saharan dust in the Houston area has been well documented (Bates et al., 2008; Bozlaker et al., 2013, 2019; Das et al., 2022), and the HYSPLIT concentration-weighted back
trajectories for this period indicate an obvious source region close to West Africa (Fig S8e). Fe$^+$, another major component Saharan dust, shows sporadic enhancements throughout the campaign and these occurrences often, but not always, coincide with elevated concentrations of K$^+$ (Fig S9). Likewise, Na$^+$ was also intermittently detected, however this element appears to be related to biomass burning aerosol as highlighted by the simultaneous increases of Na$^+$ and C$_2$H$_4$O$_2^+$ (Figure S9). While Na$^+$ has previously been identified in BB aerosol (Jahn et al., 2021), not all periods with elevated C$_2$H$_4$O$_2^+$ coincide with Na$^+$



increases. This suggests that only certain fuel types or combustion conditions emit $Na^+$. Previous SP-AMS have linked $Na^+$
with vehicle emissions (Rivellini et al., 2020), however we observe low correlations ($r^2 < 0.1$) between $Na^+$ and HOA marker
ions such as $C_4H_7^+$ and $C_4H_9^+$.

### 3.2.2 Origins of soot particle according to single particle measurements

Eight distinct BC particle types were identified by applying K-means clustering to the ETSP measurements (Fig. 5).  These
include a hydrocarbon-like OA (HOA $_{BC,ETSP}$), two rBC-rich classes displaying different amounts of BC content (rBC-1$_{ETSP}$
and rBC-2$_{ETSP}$), an oxidized OA classes (OOA $_{BC,ETSP}$), a sulfate-rich class (SO$_{4, BC,ETSP}$), a chloride-rich class (Cl $_{BC,ETSP}$), a
nitrate-rich class (NO$_{3, BC,ETSP}$) and a potassium-rich class (K$_{BC,ETSP}$).

HOA$_{BC,ETSP}$ is the most abundant particle type by number, with 3851 particle counts (26%).  The average spectrum displays
characteristic hydrocarbon peaks at $m/z$ 41, 43, 55, 57, 69 and 71 consistent with previous measurements of vehicle exhaust
(Fig 5a) (Canagaratna et al., 2004; Collier et al., 2015).  The elevated $m/z$ 44 signal indicates the aerosol has undergone some
degree of processing (Willis et al., 2016; Zhang et al., 2005).  The diurnal profile shows two increases at 6:00-8:00 and 15:00-
16:00 local time, corresponding with morning and afternoon rush hours, respectively (Figure S7e).  $C_x^+$ peaks only account for
5% of the average total mass within this aerosol type, indicating substantial coating mass.  Additionally, The HOA$_{BC,ETSP}$ class
shows a broad mass-based size distribution between 200—1000 nm in D$_{va}$. This is in contrast to other AMS measurements,
which often identify HOA with size distributions in sizes less than 150 nm (Collier et al., 2015; Lee et al., 2019; Ulbrich et al.,
2012; Zhang et al., 2005).  The larger particle size and small mass fraction of rBC observed here suggests that the particles
grouped in the HOA$_{BC,ETSP}$ class are a result of the rapid condensation of lubricating oil and other low volatility vapors onto
BC cores, or the coagulation of BC-free HOA-like particles with BC-containing particles within the vehicle exhaust system.
While previous studies have detected lubricating oil signatures in vehicle POA (Collier et al., 2015; Sonntag et al., 2012;
Worton et al., 2014), these results provide direct evidence that the smaller particles identified in these previous studies may be
externally mixed from rBC.  It is also possible that these particles may originate from other fossil fuel combustion sources,
such as emissions from heavy fuel oil used in ship traffic. Direct measurements of ultra-low sulfur diesel fuel emissions, similar
to what ships use near shore and in U.S. ports, show spectra similar to vehicle emissions (Price et al., 2017).

Two particle classes, rBC-1$_{ETSP}$ and rBC-2$_{ETSP}$, were identified, each accounting for 9% and 10% of the particle number,
respectively.  Both classes are dominated by $C_x^+$ fragments but the amount of coating material differs.  The soot particles in
the rBC-1$_{ETSP}$ class are nearly uncoated, with nearly all particles having R$_{Coat/BC}$ less than 1.  In contrast to this, those in the
rBC-2$_{ETSP}$ class have higher contributions from organics, sulfate and potassium with rBC accounting for 49% of mass on
average.  Both rBC-1$_{ETSP}$ and rBC-2$_{ETSP}$ show increased concentrations in the afternoon, concurrent with the increases in
HOA$_{ETSP}$, OOA$_{PMF}$ and O-BC$_{PMF}$.  However, unlike these other particle types, little enhancement of rBC-1$_{ETSP}$ and rBC-2$_{ETSP}$
is seen during morning rush hour.  This suggests the presence of local, non-vehicle rBC sources or meteorological conditions
conducive to the transport of fresh emissions to the sampling site during the afternoon and are likely related to industrial
emissions or gas flaring.  Direct measurements of PM emissions from gas-flaring activity observed extremely thinly coated




soot particles with BC mass fraction between 80-96% (Fortner et al., 2012). The presence of sulfate (7%) and potassium (3%) in rBC-2$_{ETSP}$ may indicate co-emission from refineries or biomass burning, respectively. The organic signal at $m/z$ 60 appears

to be elevated, further supporting the influence of BB. However, the attribution of $m/z$ 60 between C$_4^+$ and organic fragments in the ETSP analysis is uncertain. Additionally, the size distribution shifted from a peak at 200nm for rBC-1$_{ETSP}$ to 400nm for rBC-2$_{ETSP}$, consistent with the condensation of secondary material.

The average spectrum of the OOA$_{BC,ETSP}$ particle cluster was dominated by $m/z$ 44, suggesting significant oxidation of the organic material. Indeed, the fraction of mass at $m/z$ 44 ($f_{44}$) for OOA$_{BC,ETSP}$ is higher than that of OOA$_{BC,PMF}$, indicating that

particles within this cluster are more oxidized than the PMF factor. The diurnal profile of this class is relatively flat with only a modest increase in the afternoon, suggesting that OOA$_{BC,ETSP}$ represents regionally transported background aerosols that have undergone photochemical processing.

Similar to the SO$_{4,BC,PMF}$ factor, soot particles in the SO$_{4,BC,ETSP}$ class show a large contribution from H$_y$SO$_x$ fragments, with sulfate accounting for 54% of the total mass on average. The majority of the rest of the material is composed of organic species

(33%) and rBC (9%). These particles likely represent the incorporation of inorganic sulfate and other secondary organic material with rBC particles through processes such as condensation or coagulation. Given that most of the sulfate signal was attributed to this particle class, this indicates that particulate sulfate, which is primarily formed through the secondary oxidation of SO$_{2(g)}$, is generally externally mixed from organic aerosol. This class shows the largest size mode, peaking at 600 nm. As sulfate is efficiently produced in the aqueous phase, this contribution of mass in the droplet mode is consistent with aqueous

phase production. SO$_{4,\ BC,ETSP}$ consisted of thickly coated particles, with a median coating thickness of 9.5 (Fig. 2b).

The average spectrum of soot particles within the Cl$_{BC,ETSP}$ cluster is primarily composed of $m/z$ 29 and 44. However it is notable as the only particle class with measurable chloride, which accounts for 2.5% of the mass. The average cluster mass spectrum shows minor enhancements of $m/z$ 60 and 73, suggesting a potential association with oxidized BB aerosol. Particles of this type were relatively infrequent, only accounting for 5% of the measured particles.

The K$_{BC,ETSP}$ shows the highest contribution of K$^+$, accounting for 56% of the average particle mass in this cluster. The average spectrum shows minor contribution from $m/z$ 36 (C$_3^+$) and $m/z$ 44 (CO$_2^+$). This is in contrast to K$_{BC,PMF}$ which shows a significantly higher contribution from oxidized organic fragments. Recent single particle measurements of ambient biomass burning emissions revealed that some types of BBOA particles are relatively enriched with K$^+$ compared with other BBOA particle types (Lee et al., 2016). Furthermore, a near-roadside study in Fontana identified a similar K$^+$ dominated particle

cluster that the authors attributed to vehicle emissions (Lee et al., 2019). However, the Fontana K-class displayed a smaller average diameter and more signal at $m/z$ 43 and 55 than the K$_{BC,ETSP}$ cluster identified in this work. The low contribution of markers for primary HOA (i.e. $m/z$ 55, 57) and BBOA (i.e. $m/z$ 60, 73), in conjunction with the presence of secondary material such as sulfate, suggests that particles of this type have undergone atmospheric processing. From this, we conclude that the K$_{BC,ETSP}$ class represents a subset of relatively aged potassium-containing particles from sources such as vehicles, biomass

burning and/or mineral dust. This class only consisted of 499 particles (3.4% of total) with a size distribution peaking at 400-600nm.





Finally, the nitrate class was the least abundant with only 247 particles (1.5%). The average spectrum is dominated by $m/z$ 46 ($NO_2^+$) with only a small enhancement of $m/z$ 30 ($NO^+$) and $m/z$ 44. Overall, nitrate accounted for an average of 68% of particle mass in this cluster. The extremely low contribution from rBC is consistent with the overall external mixing state of nitrate
and rBC as discussed previously.

### 3.3. Sources of Refractory Metal Compounds in Soot Aerosol

An advantage of the laser vaporization utilized in the SP-AMS is the ability to detect refractory metal compounds internally mixed with rBC (Bibi et al., 2021; Cao et al., 2022; Carbone et al., 2015; Rivellini et al., 2020). Heavy fuel oil combustion from shipping activity is typically associated with the emission of heavy metals, such as vanadium (V) and nickel
(Ni) (Popovicheva et al., 2012; Rivellini et al., 2020). Indeed, a previous study concluded that shipping activities are the primary source of V-rich aerosol in the Houston region (Bozlaker et al., 2019). Two events with elevated $V^+$ concentration are detailed in Figure S10. These events coincided with increases in particulate sulfate and rBC, supporting the role of shipping related emissions, however there is negligible correlation ($r^2 < 0.05$) between $V^+$ and sulfate for the entire campaign, indicating that primary ship emissions are not the major source of sulfate for most of this campaign.

Finally, there is a noticeable increase in $K^+$ and $Fe^+$ levels on the night of Jul 4, which we attribute to firework activity. A previous SP-AMS study found enhancement of various metal species, including these ions, when fireworks were used (Bibi et al., 2021). However, the primary firework tracer identified in Bibi et al., (2021) was $Sr^+$, which was absent in our data. The lack of $Sr^+$ may be due to significantly lower concentrations during this campaign, or the utilization of different types of fireworks.

### 3.4 Black Carbon Mixing State

Figure 6a presents the distribution of single particle diversity ($D_i$) for each of the particle classes. $D_i$ is a measurement of the effective number of species present in individual species and ranges from 1 to 6 in this work. Particles in the $HOA_{BC,ETSP}$ (average $D_i = 1.5$) and $OOA_{BC,ETSP}$ (average $D_i = 1.9$) classes show low $D_i$ values as the composition is dominated by organic aerosol. Likewise, particles in the $rBC-1_{ETSP}$ class also shows low $D_i$ as they have a high mass fraction of rBC. In contrast,
$rBC-2_{ETSP}$ (average $D_i = 2.3$), $SO_{4,ETSP}$ (2.2) and $K_{ETSP}$ (2.7) display the highest diversity. All three of these classes likely consist of particles that have undergone atmospheric processing with significant contribution from secondary material such as organics and sulfate as well as primary species such as rBC and K. This highlights the role of photochemical processing in modulating the aerosol mixing state.

The mixing state index ($\chi$) indicates the degree of aerosol heterogeneity and ranges between 0% (fully externally mixed) and
100% (fully internally mixed). In this study, the hourly averages of $\chi$ range from 5% to 95%, with an average of 41 ± 13% (Fig. 7a). The bulk aerosol diversity ($D_\gamma$) calculated from the ensemble measurements exhibits a daytime decrease, indicating reduced aerosol population diversity during the day (Fig. S7n). This decrease corresponds to elevated mass fractions of organics and sulfate. In contrast to this, the single particle diversity ($D_\alpha$) shows little diurnal variation, with only a slight





decrease in the afternoon. As a result, there is an overall daytime increase in the mixing state index ($\chi$). Our results indicate
that the production of SOA and sulfate coatings on rBC through photochemical processes results in a more internally mixed,
homogenous aerosol population. Finally, at night, sulfate concentration remains elevated, while organic and rBC decrease,
resulting in an increase in $D_\gamma$ but a decrease in the mixing state index. In contrast to these results, Lee et al. (2019) reported
elevated values for both $D_\alpha$ and $D_\gamma$ during rush hour in Fontana, CA, where the rapid secondary formation of particulate nitrate
played a key role in rBC mixing state. The observed differences can be explained by the complex combination of rBC sources
in Houston and differences in aerosol acidity.

**3.5 Effect of Mixing State on Particle Hygroscopicity**

The ability for a particle to act as a CCN is governed by its bulk hygroscopicity. Previous studies have used measurements of
single particle composition to quantitatively predict their hygroscopicity. These estimates have been found to agree well with
direct measurements of water uptake using instruments such as a hygroscopic tandem differential mobility analyzer (HTDMA)
(Gysel et al., 2007; Healy et al., 2014; Petters and Kreidenweis, 2007; Wang et al., 2020b). Figure 6b shows the predicted
hygroscopicity parameter $\kappa$ for the BC-containing particles measured in this study grouped by particle classes. Details about
the calculation of $\kappa$, estimation of single particle O/C and H/C values, and the necessary assumptions are provided in S1.4.
Among the eight particle classes, the $HOA_{BC,ETSP}$ and $rBC\text{-}1_{ETSP}$ classes show the lowest hygroscopicity, with average $\kappa$ of
0.09 and 0.06, respectively. Although most particles in the $HOA_{BC,ETSP}$ class have a low rBC volume fraction, the organic
species present are chemically reduced and display extremely low $\kappa_{org}$ values, resulting in a decrease in overall particle
hygroscopicity (Fig S11). This is consistent with direct measurements of motor oil and diesel emissions, which show little
hygroscopic growth (Fofie et al., 2018; Lambe et al., 2011). Despite the presence of relatively hygroscopic coating material
on $rBC\text{-}1_{ETSP}$ and $rBC\text{-}2_{ETSP}$, the high rBC volume fraction result in extremely low overall $\kappa$ for both classes. However, the
$rBC\text{-}2_{ETSP}$ class shows slightly higher $\kappa$ values than $rBC\text{-}1_{ETSP}$ due to a higher fraction of oxidized organic material and
inorganic sulfate.
The median predicted $\kappa$ for $OOA_{BC,ETSP}$ of 0.27 agrees well with SOA produced in laboratory and chamber experiments, as
well as OOA measured in other field studies (Chang et al., 2010; Fofie et al., 2018). Additionally, due to the high mass fraction
of sulfate in $SO_{4,BC,ETSP}$ particles, the $\kappa$ value approaches that of pure ammonium sulfate (0.61) for a subpopulation of this
particle type. Similarly, the $\kappa$ value for $NO_{3,BC,ETSP}$ and $K_{BC,ETSP}$ approach that of ammonium nitrate (0.67) and potassium
sulfate (0.69), respectively. As these inorganic salts are hygroscopic, the overall hygroscopicity of these particles are highest.

**4. Conclusions**

This study involved the measurements of both the bulk and single particle composition of soot particles using a SP-AMS
operated in the laser-only configuration. Material internally mixed with BC had a significant contribution from secondary
aerosol species, including oxidized organic aerosol and inorganic sulfate. This finding highlights the importance of



atmospheric aging processes affecting soot aerosol characteristics in urban environments. The utilization of two different source apportionment techniques in this study, namely PMF of the ensemble measurements and K-Means clustering of the single particle measurements, yields complimentary results. Although PMF analysis of the ensemble spectra failed to isolate a factor related to HOA, either due to low signal intensity or a similar temporal profile as other OA sources, the simultaneous application of the ETSP mode successfully identified particles exhibiting HOA-like spectra. Additionally, both source

apportionment techniques revealed the presence of nearly pure rBC particles with extremely small amounts of coating material. For instance, the average $R_{coat/BC}$ for the oxidized rBC factor (O-BC$_{PMF}$) was only 0.24 while the median value for the rBC-1$_{ETSP}$ single particle cluster was 0.20.

A significant limitation in the ETSP data interpretation is the reliance on UMR mass spectra due to the extremely low ion counts for individual particle events. This introduces uncertainty when distinguishing different aerosol species that share

isobaric ions, such as $C_4^+$ and $SO^+$ or $C_5^+$ and $C_2H_4O_2^+$. Additionally, the lack of high-resolution spectra hinders the differentiation of isobaric ions representing distinct organic ion families, such as $C_4H_7^+$ and $C_3H_3O^+$ at *m/z* 55, which would otherwise contribute to more robust differentiation between POA and SOA factors, such as HOA and OOA.

The utilization of single particle measurements also allowed for a detailed analysis of the BC mixing state. The average mixing state index ($\chi$) was determined to be $41 \pm 13\%$, indicating that the aerosol population demonstrated an intermediate between

internal and external mixing. Notably, these values align with the $\chi$ values measured at two other cities, including a near roadside location in Fontana, CA (48%) and a downtown location in Pittsburgh, PA (43%) (Lee et al., 2019; Ye et al., 2018). $\kappa$-Köhler theory was utilized to estimate critical supersaturation values at which these particles would activate into cloud droplets (Fig. 7b). Many of the BC-containing particles had undergone atmospheric processing and exhibited mixing with hygroscopic material. Consequently, a significant fraction of the soot particles demonstrates sufficiently high $\kappa$ values and

diameters to activate under atmospherically relevant supersaturation values. Among the classes identified, the OOA$_{BC,ETSP}$ and SO$_{4, BC,ETSP}$ type particles appear to be particularly relevant for CCN activity, given their combined attributes of abundance, hygroscopicity and size distribution. This finding has important implications for understanding the activation of rBC containing particles into cloud droplets and the sources of CCN in urban locations.

**Acknowledgement**

This research was supported by the U.S. Department of Energy's Atmospheric System Research Program (Grant # DE-SC0021242 and DE-SC0022140), and the California Agricultural Experiment Station (Project CA-D-ETX-2102-H). R.F. also acknowledges funding from the Jastro-Shields Research Award and the Donald G. Crosby Fellowship in Environmental Chemistry from the University of California at Davis.



## 515 Competing Interests

At least one of the (co-)authors is a member of the editorial board of ACP.




**Figure 1: (a) diurnal profile of rBC concentration, coating to rBC mass ratio ($R_{coat/BC}$) and gas-phase $NO_x$. (b-e) diurnal profile of aerosol species mixed with BC (colored traces with markers) and present in the NR-PM$_{2.5}$ fraction (black traces). (e) Correlation between rBC measured by SP-AMS and co-located SP2. (f-h) scatterplot between species concentration present in BC fraction and NR-PM$_{2.5}$ fraction. Trend lines are orthogonal distance regression. All units are in μg m$^{-3}$, except for $R_{Coat/BC}$, which is a**
**dimensionless value.**







**Figure 2: (a) normalized frequency of $R_{coat/BC}$ from the ensemble and ETSP measurements. (b) Frequency of each single particle class as a function of $R_{Coat/BC}$. (c) Average mass fraction of ensemble measurements, divided by $R_{coat/BC}$, and the O/C and H/C ratios of the organic coating. (d) Average mass fraction of single particle measurements, divided by $R_{coat/BC}$.**





**Figure 3: Average fractional composition of total PM₁,ʙᴄ measured by (a) high-resolution ensemble measurements and (b) ETSP measurements. (c) Average mass spectrum of ensemble measurements and (d) average mass spectrum of all single particle events measured by ETSP mode. (e-j) Comparison of species dependent size distributions for ensemble measurements (colored line) and ETSP measurements (black line). Bulk size distributions are mass weighted (against right y axes), while ETSP measurements are signal weighted (against left y axes).**



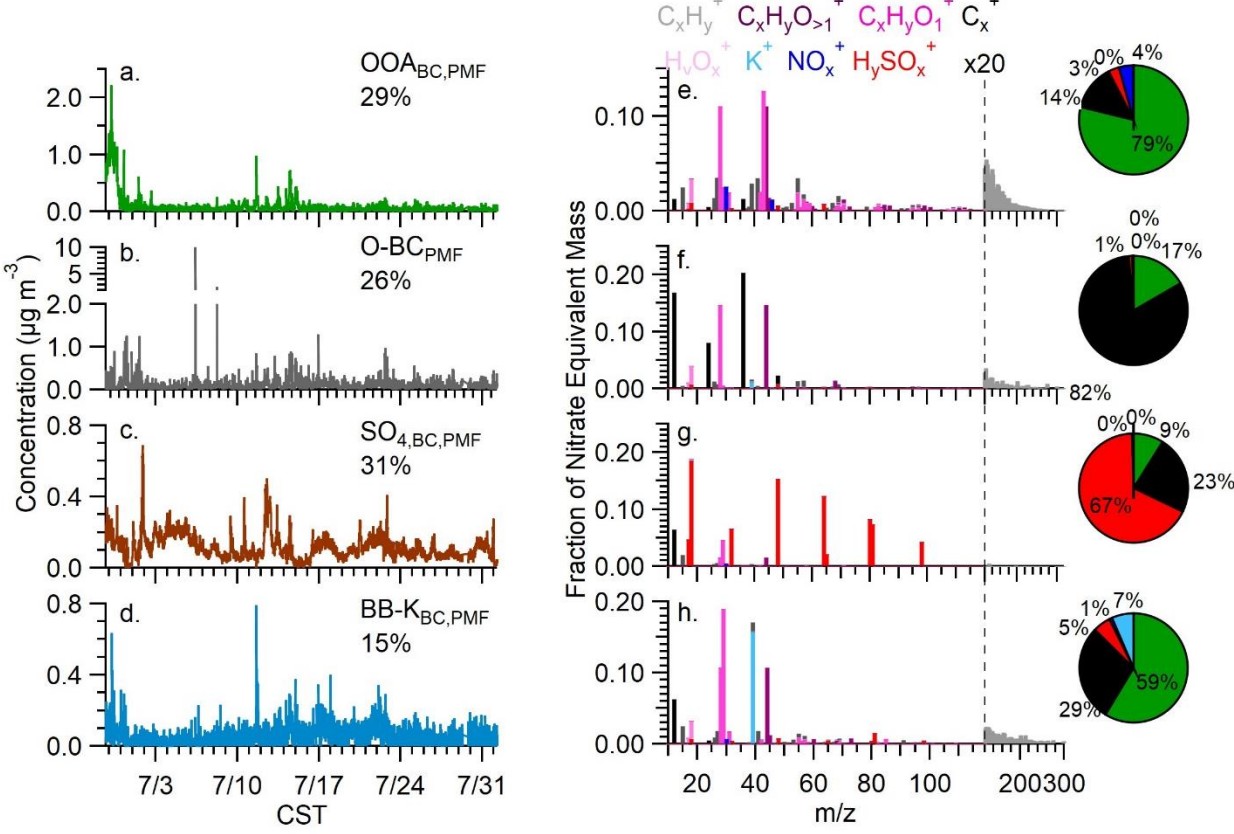

**Figure 4: (a-d)** Concentration time series of the PMF factors. Values reported are percent of total PM$_{1,BC}$. **(e-h)** Normalized, nitrate-equivalent mass spectra of PMF factors. Pie charts show the mass fraction of each species after applying species depedent RIE.



**Figure 5: (a-h)** Average mass spectra of each ETSP cluster calculated using K-means clustering. Number of particles in each class is also displayed. **(i-p)** Signal weighted size distribution of each cluster. Pie charts display the mass fraction of each aerosol species.



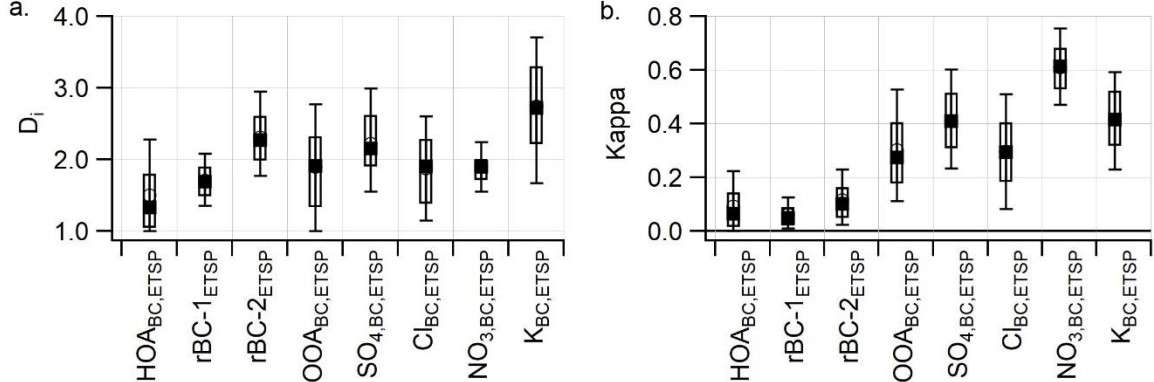

**Figure 6: (a)** Single particle diversity (Dᵢ) for each cluster type. **(b) Predicted hygroscopicity for individual particles, divided by particle class.** The solid and open markers indicate the median and mean respectively, the box indicates the 25th-75th percentile, and whiskers indicate 10th-90th percentiles.



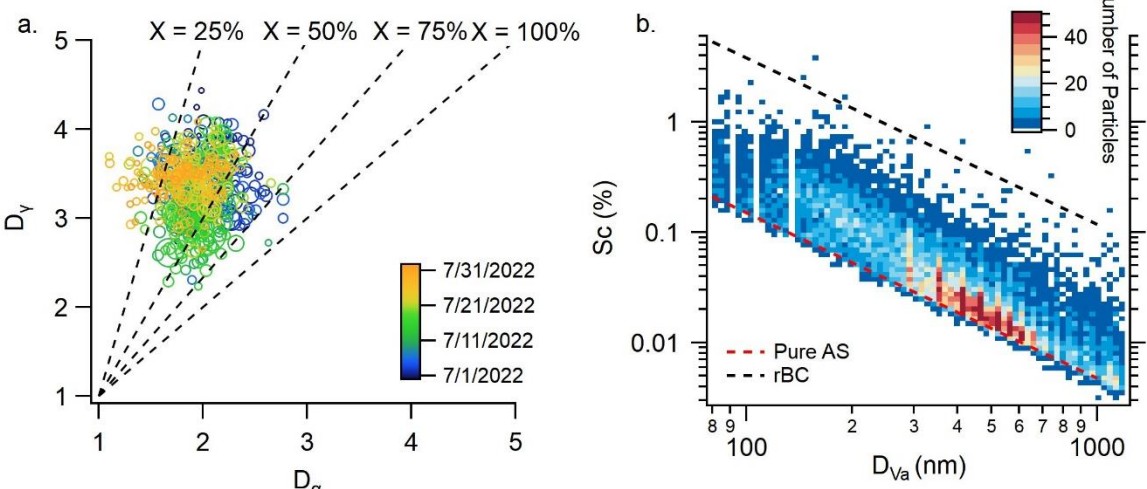

**Figure 7: (a) Mixing state diagram depicting the relationship between bulk particle diversity ($D_\gamma$), average single particle diversity ($D_\alpha$). The dashed lines indicate the resulting mixing state index ($\chi$). Points are sized by $PM_{1,rBC}$ concentration and colored by sampling date. (b) Bivariate histogram of estimated particle critical supersaturation and particle diameter. Dashed lines represent uncoated rBC ($\kappa = 0.001$) and pure ammonium sulfate ($\kappa = 0.61$).**



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
