# Peer review of "Chemical Properties and Single Particle Mixing State of Soot Aerosol in Houston during the TRACER Campaign"

_EGUsphere, 2023_

## Author Comment (AC1)

**Response to reviewers**

We thank the reviewers for their thoughtful and valuable comments, and we have incorporated their suggestions into the revised manuscript. Listed below are our point-by-point responses to the comments and the corresponding manuscript revisions in blue. The review comments are replicated in italic.

**Reviewer 1**

*General Comments:*

*Overall this paper is very well written and gives a good description of the application of a new technique to gain knowledge with respect to the speciation of black carbon containing aerosols. I would recommend this paper for inclusion to this journal after some minor revisions and other points are considered. Points are listed below.*

*Line 103: Vmode is the standard default mode of operation for any field measurements with AMS. This is just a wording thing but when you state cost of lower resolution it makes it sound as if you are running the instrument in sub optimal which you really aren't. I know W mode gives you 2\*resolution in V mode but at about 1/10 the sensitivity so V mode is the typical default mode.*

We have rephrased the sentence in question as V-mode is the typical default setting for ambient measurement. It now reads:

"The mass spectrometer was operated in the optical "V" mode with a mass resolution ($m/\Delta m$) of 2500 and was programmed to switch between three different sampling modes: the ensemble mass spectrum (MS) mode, the efficient particle time of flight (ePToF) size mode, and the event trigger single particle (ETSP) mode"

*Line 319; the thickly coated particles could also be from a dirty combustion source like BB or cooking which is heavily coated on primary emission.*

We think the reviewer's comment meant to refer to Line 219. It is an important point that thickly coated rBC can be directly emitted from primary sources, and it is possible that periods of high coating thickness are due to these sources. However, previous work has shown that cooking aerosol is unlikely to be internally mixed with black carbon (i.e. (Lee et al., 2017)) The sentence now reads:

"However, there were periods of thickly coated particles with $R_{Coat/rBC}$ values exceeding 10, indicating that a portion of the soot aerosol at this site may have undergone extensive atmospheric processing or were emitted with an existing coating by a primary source such as biomass burning"

*Line 234; can you get insight to the SO4 makeup by looking at the conventional AMS data as well*

Although it is possible of extracting additional information on sources of sulfate in the region from the data obtained by the collocated dual vaporizer SP-AMS, conducting a detailed analysis to examine the

the mixing states of sulfate aerosol is beyond the scope of this work. Furthermore, it is likely that the sources of externally mixed sulfate are significantly different than the sulfate incorporated in rBC aerosol.

*Line 243; I understand the need for both of these assumptions. Internally mixed within BC wont vap with conventional heater and the instruments had different size cut lenses Standard vs 2.5 Can you give any information like size profile with SMPS or just theory to back up these assumptions.*

To help validate the assumption regarding the difference in instrument size cuts, the dual vaporizer SP-AMS size distributions were integrated to determine the fraction of mass <1 μm and between 1-2.5 μm. We found that 72% of sulfate, 69% of OA and more than 90% of nitrate was present in the $PM_1$ fraction. We have edited the section to read:

The comparison of each species between the laser-only SP-AMS and the dual vaporizer SP-AMS, operated in laser-off mode, provides valuable insights into the percentage of mass internally mixed with rBC. This comparison assumes that the mass of refractory organics, sulfate, nitrate, and chloride material internally mixed with BC is low. To estimate the non-refractory mass of each species present between 1-2.5 μm, we used the size distributions measured by the collocated dual vaporizer SP-AMS. From this, it was found that 72% of sulfate, 69% of organic aerosol, and more than 90% of nitrate was present in the $PM_1$ fraction. When accounting for the instrument's size range difference, approximately 19% of submicron organic mass during this study is internally mixed with rBC. However, this fraction exhibited considerable temporal variation, as indicated by the moderate $r^2$ of 0.49 (Fig. 1f).

*Line 273 good point mz 36 bias*
*Line 279 good point on larger particles more likely to trigger*
*Line 286 good point on some particles having poor overlap and therefore only partial vap.*

We thank the reviewer for their positive feedback.

*Lines 340 – 360 Are any of these metals being seen without BC? I know the laser does vaporize many elemental metals. If so could help differentiate between BB and dust.*

Although it would be useful to determine if some metals were detected without the presence of rBC, the mass spectra used for PMF analysis are five-minute averages. Thus, we are not able to conclude if the metals are internally mixed with rBC. Additionally, all single particle spectra are unit mass resolution making the detection of metals uncertain in the event trigger measurements.

*Lines 370 – 380 Could this also be due to not enough ions with smaller particles to trigger an event.*

This is certainly a possibility, the following sentence has been added

"Another possibility is that smaller HOA particles generate fewer ions and are less likely to trigger an event as discussed in Section 3.1.2"

*Line 390 Should be C5+*

The typo has been fixed

*Line 435 – 439 Again is any of this metal without BC?*

As mentioned in the previous response, with the current instrumental setup we are unfortunately unable to determine whether the metals are mixed with rBC.

*Lines 495 497 Good point on the isobaric issues.*

We thank the reviewer for their positive feedback.

*Figure 1 a-d I assume the shading is error bars for the SP-AMS measurements but I don't see this stated anywhere.*

The shaded area is the interquartile range of the measurements.  This was added to the caption which now includes:

(a) diurnal profile of rBC concentration, coating to rBC mass ratio ($R_{coat/BC}$) and gas-phase $NO_x$. (b-e) diurnal profile of aerosol species mixed with BC (colored traces with markers) and present in the NR-PM$_{2.5}$ fraction (black traces). Markers indicate the median and shaded area indicates the interquartile range. (e) Correlation between rBC measured by SP-AMS and co-located SP2. (f-h) scatterplot between species concentration present in BC fraction and NR-PM$_{2.5}$ fraction.  Trend lines are orthogonal distance regression. All units are in $\mu g\ m^{-3}$, except for $R_{Coat/BC,}$ which is a dimensionless value.

*Figure 3 e-j This highlights the issue of smaller sizes not being detected as well with event trigger.*

Yes, we agree. Event trigger does tend to exhibit bias against smaller particles.

**Reviewer 2**

*This paper investigates chemical properties and mixing state of BC-containing particles by using both Aerodyne laser-only SP-AMS and single particle AMS during a field campaign, It provides a very valuable dataset that allows comprehensive analysis on the BC aerosol properties, the paper is very well written and the interpretation of the observed results is robust and trustworthy, the paper can be accepted after a minor revision, the questions are listed below:*

We thank the reviewer for their encouraging comments.

*line285: Did you remove the tungsten in the ETSP mode?*

Yes, the ions influenced by tungsten ions *m/z* 182 ($^{182}W^+$), 183 ($^{183}W^+$) and 186 ($^{186}W^+$) were removed prior to the clustering analysis. This is included at line 145.

*line310: The figure of back trajectories should point to Fig. S8b.*

This has been corrected.

*line95: I think it would be more interesting to present a comparison of the composition between rBC-PM1 and NR-PM1~2.5, as well as between rBC-PM1 and NR-PM1 separately in Fig.1.*

To provide a more accurate comparison between the two instruments, the fraction of NR-PM was estimated from the size distributions from the dual vaporizer SP-AMS.  This is detailed in the response to the previous reviewer, and on lines 244-248.

*Have you assessed the impact of BC-free particles in ensemble measurements? It is noteworthy that they exhibit a higher presence in organic components at m/z 29, yet a lower presence at 60 and 73 in the ensemble compared to those in ETSP (Figures 3c and 3e).*

The single particle measurements presented in this work highlight the potential influence of BC-free particles on the ensemble measurements, however the impact of this on the ensemble mass spectra is difficult to quantify.  We also agree that there are subtle differences between the average ensemble and ETSP organic spectra which were not discussed in the text.  The following was added (Line 278-280):

"Additionally, the average single particle OA composition shows a slight enhancement of certain fragments, including *m/z* 60 and 73, two markers for biomass burning organic aerosol (BBOA; Cubison et al., 2011).  This suggests there may a positive bias towards certain sources, such as BBOA."